# Stratification in planetary cores by liquid immiscibility in Fe-S-H

Shunpei Yokoo [1✉], Kei Hirose [1,2], Shoh Tagawa [1,2], Guillaume Morard[3] & Yasuo Ohishi[4]

Liquid-liquid immiscibility has been widely observed in iron alloy systems at ambient pressure and is important for the structure and dynamics in iron cores of rocky planets. While such previously known liquid immiscibility has been demonstrated to disappear at relatively low pressures, here we report immiscible S(±Si,O)-rich liquid and H(±C)-rich liquid above ~20 GPa, corresponding to conditions of the Martian core. Mars' cosmochemically estimated core composition is likely in the miscibility gap, and the separation of two immiscible liquids could have driven core convection and stable stratification, which explains the formation and termination of the Martian planetary magnetic field. In addition, we observed liquid immiscibility in Fe-S-H(±Si,O,C) at least to 118 GPa, suggesting that it can occur in the Earth's topmost outer core and form a low-velocity layer below the core-mantle boundary.

[1] Department of Earth and Planetary Science, The University of Tokyo, Tokyo, Japan. [2] Earth-Life Science Institute, Tokyo Institute of Technology, Tokyo, Japan. [3] Université Grenoble Alpes, Université Savoie Mont Blanc, CNRS, IRD, Université Gustave-Eiffel, ISTerre, 38000 Grenoble, France. [4] Japan Synchrotron Radiation Research Institute, SPring-8, Sayo, Japan. ✉email: shunpei@eps.s.u-tokyo.ac.jp

**M**etallic iron cores of terrestrial planets and moons likely contain more than one light elements[1,2]. Multiple light elements in molten iron often cause liquid–liquid immiscibility at 1 bar and high pressure such as in Fe-O, Fe-S-O, Fe-S-Si, and Fe-S-C[3–9]. The liquid immiscibility in planet/satellite iron cores gives rise to their stable stratification. At the same time, the density difference between immiscible two liquids can drive core convection, leading to a planetary dynamo action and magnetic field generation for a period of time. Previous high-pressure and -temperature (P-T) experiments, however, revealed that a miscibility gap closes below 10–30 GPa in all of these iron alloy systems mentioned above[3–9]. Thermodynamic modeling suggested that the Fe-S-O system exhibits an extensive liquid immiscibility field even under core pressures[10], but it is not supported by high-pressure experiments[5,11]. Also, while two immiscible liquids were recently reported in Fe-Si-O to 140 GPa[12], it was not reproduced by theoretical calculations[13] nor found in earlier experiments[14].

Here, we report the immiscibility between liquids with Fe-S(±Si/O) and Fe-H(±C), based on melting experiments in a diamond-anvil cell (DAC) to 118 GPa, close to the pressure range of the Earth's core. Sulfur may be commonly present in planetary cores as it is frequently included in iron meteorites. It is important, in particular for the Martian core, because chalcophile (sulfur-loving) elements are remarkably depleted in its silicate part[15]. Recent planet formation theories demonstrate that a large amount of water was delivered to both Mars and the Earth during their accretions[16,17], suggesting that hydrogen is possibly a major light element in the core[18–20]. Despite its importance, so far the Fe-S-H system has been little investigated at high pressures[21]. We discuss implications of the immiscibility in liquid Fe-S-H for structure, dynamics, and the formation of a magnetic field in planetary iron cores.

## Results and discussion

**Liquid immiscibility between Fe-S and Fe-H alloys.** We performed melting experiments on the Fe-S-H ± C/Si/O system in a wide P-T range (20–118 GPa, 1860–3700 K) and found a homogeneous single liquid or two separate immiscible Fe-S(±Si/O) and Fe-H(±C) liquids (Table 1). Several different, liquid-liquid immiscibility textures were observed. When liquids are relatively enriched in carbon (1.6–5.9 wt% C), the textures of the two immiscible liquids are very similar to those observed in previous experiments with a large-volume (multi-anvil) press conducted below ~20 GPa[5,6] (Fig. 1a). With <~1 wt% carbon concentration, immiscible Fe-S and Fe-H liquids coexist with a clear boundary as shown in Fig. 1b—the S-poor liquid contains more bubbles and cracks, indicating the presence of a higher concentration of hydrogen that escaped during decompression[22,23]. These Fe-S and Fe-H liquids were found to be miscible at relatively high temperatures (>2000 K at ~20 GPa and >~3000 K at higher pressures) (Fig. 1c and Table 1).

Some experiments showed a change in texture of liquid according to temperature variations in a laser-heated sample. In run #6 (Fig. 1d), we found coexisting immiscible two liquids in a relatively low-temperature region, which are detached from a homogeneous liquid Fe-S-H pool in the hottest part. On the other hand, in the case of a single liquid pool (run #4, Fig. 1e), the sulfur concentration in the liquid gradually decreases toward the outside and sharply increases at the edge, which also indicates the change from one liquid to two immiscible liquids with decreasing temperature. We also observed a separation of two immiscible liquids at <300 nm scale in a relatively low-temperature part, while their bulk composition is identical to that of a homogeneous portion at the center of a heated area (runs #1 & 2, Fig. 1f). Such microscopic separation is likely

**Table 1 Experimental conditions and results.**

| Run# | Pressure (GPa) | Temperature (K) | S (wt%) | H (wt%) | C (wt%) | Si/O (wt%) |
|---|---|---|---|---|---|---|
| **Miscible** | | | | | | |
| 1 | 20 (2) | 2210–2160[a] | 7.09 (11) | 1.67 (18) | 2.1 (2) | – |
| 2 | 24 (2) | 2770–2530[a] | 15.5 (4) | 1.17 (10) | 0.5 (1) | – |
| 4 | 38 (4) | 3510–3150[a] | 16.6 (3)-12.6 (2) | – | 0.9 (1) | – |
| 5 | 39 (4) | 3490–3450 | 12.3 (2) | 1.33 (14) | 0.6 (1) | – |
| 6_1 | 40 (4) | 3620–3200 | 5.1 (4) | 1.78 (19) | 1.2 (1) | – |
| 12 | 44 (4) | 3700–3630 | 8.2 (5) | 1.52 (16) | 0.4 (1) | – |
| 13 | 53 (5) | 3350–3150 | 14.4 (2) | – | 0.3 (1) | – |
| 14 | 57 (6) | 3400–3120 | 12.2 (2) | – | 0.1 (1) | – |
| 15 | 99 (10) | 3570–3480 | 13.3 (2) | – | 0.4 (4) | – |
| **Immiscible** | | | | | | |
| 3 | 27 (3) | 1990–1860 | 16.8 (5) | 0.4 (3) | 1.1 (1) | – |
| | | | 0.013 (8) | 1.87 (19) | 1.2 (2) | – |
| 6_2 | 40 (4) | 2690–2590 | 18.4 (3) | 0.2 (2) | 0.4 (1) | – |
| | | | 0.04 (3) | 1.75[b] | 0.3 (1) | – |
| 7 | 71 (7) | 3610–3210 | 13.30 (9) | 0.2 (2) | 0.2 (1) | – |
| | | | 1.5 (5) | 1.75[b] | 0.4 (1) | – |
| 8 | 100 (10) | 3010 | 19.3 (6) | – | 1.6 (1) | – |
| | | | 0.38 (3) | 0.67 (13) | 5.9 (2) | – |
| 9 | 52 (5) | 2750 | 7.1 (2) | 1.3 (4) | 0.7 (1) | Si 2.05 (4) |
| | | | 0.004 (6) | 1.86 (23) | 0.5 (1) | Si 0.19 (2) |
| 10 | 118 (12) | >3000[c] | 7.7 (2) | – | 0.8 (2) | Si 2.6 (3) |
| | | | 0.33 (16) | 0.91 (18) | 4.4 (3) | Si 0.66 (6) |
| 11 | 32 (3) | 2050 | 2.24 (16) | 1.6 (4) | 1.4 (2) | O 0.7 (3) |
| | | | 0.02 (3) | 2.02 (23) | 1.5 (2) | O 0.02 (3) |

Numbers in parentheses indicate errors (1σ) in the last digits. Reported temperatures indicate the highest (center) – the lowest temperatures (edge), or only the highest temperature.
[a]The lower bound corresponds to the temperature at the miscible/immiscible boundary.
[b]Hydrogen contents are calculated by assuming $x = 1$ in $FeH_x$ because the X-ray diffraction from the sample was not obtained.
[c]Temperature measurement failed.

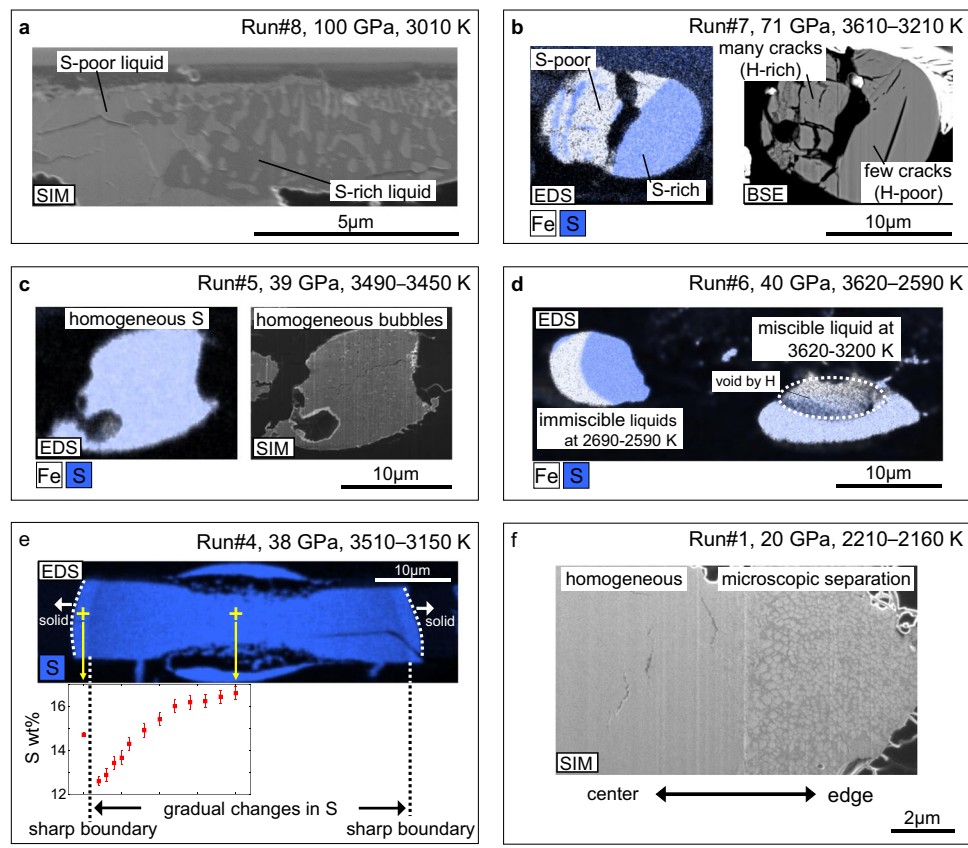

**Fig. 1 Textures of two immiscible liquids and a single homogeneous liquid.** Scanning ion microscope (SIM) and backscattered electron (BSE) images, and X-ray maps for Fe and S concentrations obtained by energy-dispersive X-ray spectrometry (EDS) for sample cross sections from runs #8 (**a**), #7 (**b**), #5 (**c**), #6 (**d**), #4 (**e**), and #1 (**f**). A sulfur concentration profile is given in **e**. Dark domains in X-ray maps represent $Al_2O_3$ pressure medium or diamond/hydrocarbon.

caused by small temperature fluctuations during laser heating. The conditions of these three experiments should be close to the boundary between one homogeneous liquid and two immiscible liquids (Fig. 2).

The experiments on Si- or O-bearing samples demonstrate that when immiscible S-rich and H-rich liquids coexist, both silicon and oxygen are preferentially incorporated into the former (Supplementary Fig. 1). On the other hand, carbon is included more in the latter above 71 GPa, while it is partitioned almost equally at lower pressures (Table 1). These relations suggest that liquid Fe alloys can be classified into two groups, Fe-S-Si-O and Fe-H-C, based on the affinity among these light impurity elements in the Earth's core pressure range. Such affinity might originate from the mechanism of incorporating light elements in liquid Fe; hydrogen and carbon have smaller atomic radii and therefore stronger interstitial characters than sulfur, silicon, and oxygen (Umemoto and Hirose[18]).

These experiments constrain the P-T conditions where two immiscible liquids are formed from a single homogeneous Fe + 7–16 wt% S + 1.2–1.7 wt% H liquid (Fig. 2a). The miscible/immiscible boundary may be fitted by T (K) = −33,400(3300)/P(GPa) + 3870(110), considering that the excess free energy arising from mixing Fe-S and Fe-H liquids includes a P-dependent term proportional to 1/P and a T-dependent term proportional to T. The boundary has a relatively steep increase in T below ~60 GPa. The two coexisting immiscible liquids contain 13–19 wt% S + 0.2–0.4 wt% H and 0–1.5 wt% S + 0.7–1.9 wt% H, respectively, indicating a wide miscibility gap between S-rich and H-rich liquids (Table 1). On the other hand, when silicon or oxygen is involved, a gap in sulfur concentration between immiscible S-rich and H-rich liquids is much smaller (2–8 wt% and 0–0.3 wt% S, respectively).

**Implications for Martian core**. The present-day P-T conditions[24,25] of the Martian core are fully within those required for liquid immiscibility to occur in the Fe-S-H system (Fig. 2b). A recent cosmochemical study of Martian meteorites best estimated the Mars' core composition to be Fe + 6.6% S + 0.9% H + 5.2% O by weight[26], which is most likely within a miscibility gap between Fe-S-O and Fe-H liquids. The initial Martian core-mantle boundary (CMB) temperature has been estimated to be 2400–3400 K at ~20 GPa[27–29], suggesting that the liquid immiscibility and resulting compositional stratification may have started from the beginning or early history of Mars.

Fig. 2b illustrates that the separation of two immiscible liquids first appeared in the deep core of Mars, because the dT/dP slope of the miscible/immiscible boundary is steeper than that for the Martian core temperature profile[24,25] at ~20–40 GPa. While separated denser liquids stayed at the deepest part, lighter liquids migrated upward and mixed with the bulk liquid core, which could drive Martian core convection (Fig. 3). At the same time, gravitationally stable, compositional stratification should have developed in a region where liquid separation took place. Eventually, Mars' entire core became stratified, which ceased convection. It is known that Mars' planetary magnetic field was present until ~4 gigayears ago and was then lost[30]. The separation of immiscible S-rich and H-rich liquids could have been responsible for both the onset and termination of Martian core convection and dynamo action. Observing the structure of the Martian core, possibly via the ongoing InSight mission[31], will validate this scenario.

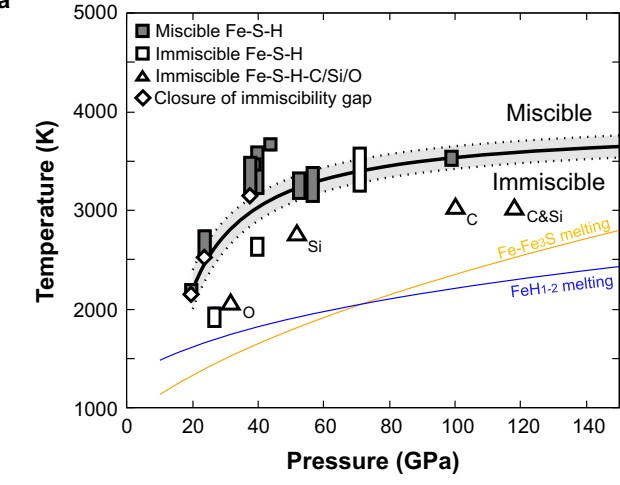

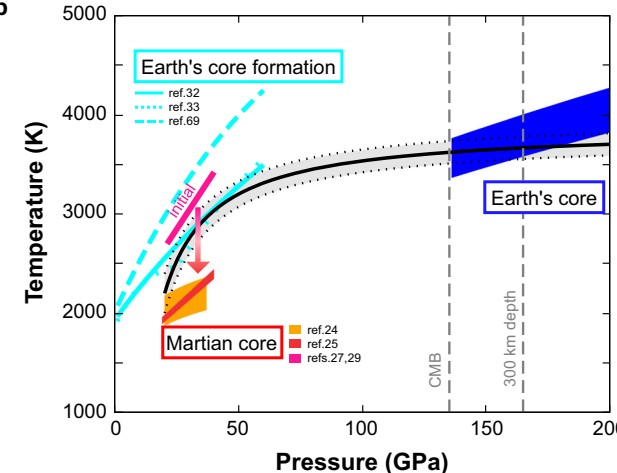

**Fig. 2 Pressure(*P*)-temperature(*T*) conditions for immiscibility in liquid Fe-S-H. a** Experimental results showing a homogeneous single liquid (filled) and two immiscible liquids (open) in Fe-S-H (rectangles) and Fe-S-H-C/Si/O (triangles). Elements other than Fe, S, and H are indicated. Miscible/immiscible boundary (black line) and its uncertainty band (dashed lines) are based on the *P-T* conditions of miscible/immiscible border in a sample (diamonds) (runs #1, 2, and 4) and those of miscible/immiscible liquids obtained near the boundary. Yellow and blue lines are melting curves in the Fe–Fe₃S[55] and Fe–FeH$_x$ (*x* > 1) systems[57], respectively. **b** Blue band indicates the isentropic temperature profile of the Earth's core with the core-mantle boundary temperature $T_{CMB} = 3600$ K[68]. Light-blue lines are the *P-T* paths for segregation of core-forming metals from silicate[32,33,69]. Temperature profiles of the initial[27,29] (pink) and present-day Martian core (orange[24] and red[25]) are also given.

**Implications for Earth's core**. Both sulfur and hydrogen could be important light elements in the Earth's core as well[18], and the *P-T* region for the Fe-S-H liquid immiscibility suggests that it could occur during core formation[32,33] (Fig. 2b). The separation of S-rich and H-rich immiscible liquids could have caused complex elemental partitioning between core-forming metals and silicate[34]. In addition, it left more sulfur in a magma ocean because H-rich liquids contain the least sulfur. If core-forming metal separated into half S-rich and half H-rich liquids, the amount of sulfur left in silicate melt could be doubled. As a consequence, more sulfide may have segregated from silicate after core formation[35], which can be responsible for isotopic signatures in some chalcophile elements[36,37] and non-chondritic abundance ratios of highly siderophile elements in the mantle[34,38]. A recent

model[39] of heterogeneous Earth accretion and multi-stage core formation argued that the mantle sulfur concentration is explained without sulfide segregation if sulfur was supplied only in the late stage of planet growth. However, it is likely that hydrogen was transported to the Earth along with sulfur[17], possibly leading to the Fe-S-H liquid immiscibility and the sulfide segregation.

Liquid immiscibility may develop also at present in the uppermost outer core (Fig. 2b). Seismology shows a low-velocity layer at the top of the outer core, recently called the E′ layer[40–43]. The E′ layer is likely to have a low-density anomaly, albeit recent observations suggest a marginally higher density[44]. The lower velocity is not explained by the simple addition of light element(s) since alloying more of a given light element causes higher sound velocity[18,45]. Chemical reaction of core metal with an FeO-rich basal magma ocean (BMO)[46] results in the depletion in Si and the enrichment in O, which can produce a light and slow layer at the top of the core[45], but the limited diffusivities of these elements may prohibit the formation of ~300-km thick E′ layer[47]. Furthermore, the BMO may have crystallized bridgmanite that floats above[48] and then dense ferropericlase[49,50] that forms a layer separating the BMO and the core and suppresses the chemical reaction between them. Alternatively, the E′ layer could be formed by Fe-S-H liquid immiscibility.

Unlike the case of the Martian core, secular cooling can cause liquid immiscibility from the top of the core (Fig. 2b). While the lighter H-rich liquid stays at the top, the denser S-rich liquid sinks and mixes with the rest of the liquid core (bulk outer core, BOC) (Fig. 3). Such H-rich liquids have accumulated to form the compositionally zoned E′ layer, with a lighter liquid at a shallower depth which has expelled more of dense liquid. Such compositional gradient may correspond to the velocity gradient observed in the E′ layer[41–43].

The light impurity component in the BOC is likely a mixture of S, H, Si, and O. With cosmo-/geochemically proposed 2 wt% S (Dreibus and Palme[51]), the remaining H, Si, and O contents are a trade-off, but their concentrations are constrained to be consistent with the observations[52] of the density and velocity of the outer core[18] (see Methods section). On the other hand, the E′ layer could be depleted in S, Si, and O but instead enriched in H and C compared to the BOC. Indeed, such compositional difference causes the low-velocity anomaly in the E′ layer (Fig. 4), because the effects of hydrogen on increasing liquid Fe velocity with respect to decreasing its density is smaller than those of other light elements and thus replacing hydrogen for other elements without changing density results in lower velocity. Seismology[41–43] shows that the low-velocity anomaly and causing compositional difference are the largest at the CMB (top of the E′ layer). The present experiments suggest that the E′ layer at its top contains no S and 40% of the Si and O contents in the BOC (Table 1, see Methods section). When the E′ layer has no density anomaly and is carbon-free, the lower velocity by 0.035 km/s[41–43] at CMB gives <0.4 wt% H in the BOC and 0.5–0.8 wt% H at the CMB (Fig. 4a), supporting that hydrogen is an important core light element[18–20,53]. Such possible range of the BOC composition is consistent with a recent core formation model[20] which suggests that 0.3–0.6 wt% H was incorporated into the Earth's core-forming metals. On the other hand, with 0.1 g/cm³ smaller density in the E′ layer than in the BOC when compared at the CMB, the H contents are < 0.2 wt% in the BOC and ~0.7 wt% at the CMB (Supplementary Fig. 2). In addition, the C/H molar ratio should be <0.1 in the E′ layer, suggesting a negligible amount of carbon in the BOC because carbon is preferentially incorporated into the H-rich/S-poor liquid as demonstrated in run #10 at 118 GPa (Table 1).

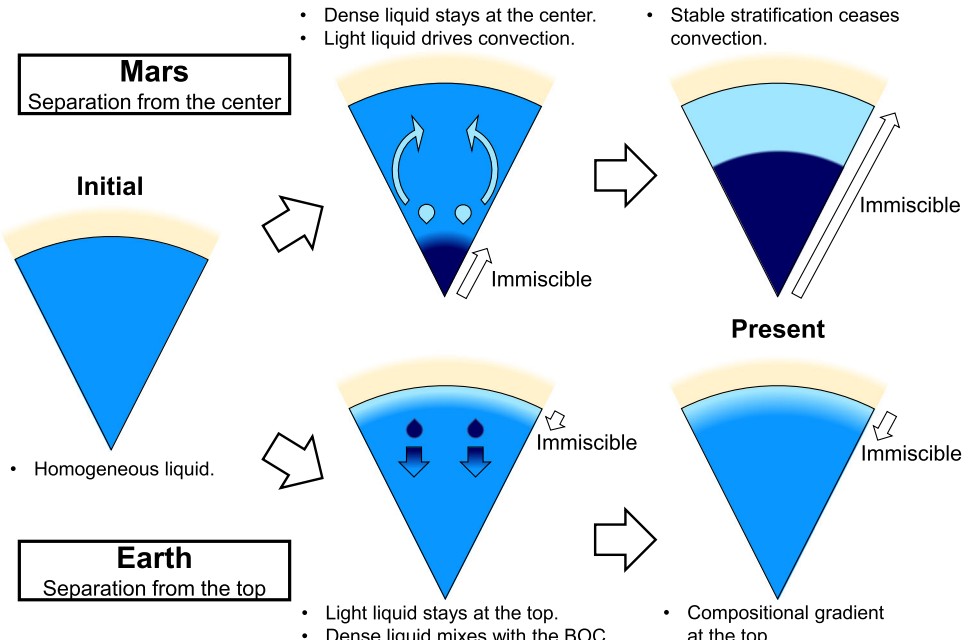

**Fig. 3 Stratification in Mars' and Earth's cores caused by liquid immiscibility.** Light- and dark-blue represent buoyant and dense liquids, respectively. In the Martian core, liquid immiscibility that started from the center had driven convection and dynamo but eventually formed entire core stratification which ended Mars' planetary magnetic field. In the case of the Earth, on the other hand, the core liquid immiscibility commenced from the top, leading to stratification at the uppermost core which is now observed as E′ layer. These illustrations are not to scale.

## Methods

**High *P-T* experiments**. We used a laser-heated DAC to generate high *P-T* conditions relevant to the cores of Mars[24,25] and Earth. The diamond anvils had flat or beveled culets with diameters of 300 or 120 μm, respectively. Starting materials were homogeneous foils of Fe-S, Fe-S-Si, and Fe-S-O (Supplementary Table 1); Fe-10.3 wt%S and Fe-2.1 wt%S-2.5 wt%Si prepared by an ultra-rapid quench method[54] and Fe-6.5 wt%S-5.2 wt%O by sputtering[14]. These samples are the same as those used in previous studies[11,55,56]. The sample was loaded into a hole at the center of a pre-indented rhenium gasket, together with $C_nH_{2n+2}$ paraffin that was used as a source of hydrogen and carbon. Note that the incorporation of carbon into liquid iron is limited in the presence of hydrogen[57] (Table 1). In some runs, iron alloy + paraffin was sandwiched by $Al_2O_3$, which served as thermal insulation layers.

After compression, the sample was heated from both sides with a couple of 100 W single-mode Yb fiber lasers at the beamline BL10XU, SPring-8, and at the University of Tokyo (runs #13-15) with and without X-ray diffraction (XRD) measurements (see below), respectively. The radial temperature gradient was diminished by beam-shaping optics that converts a Gaussian laser beam to one with a flat energy distribution. The laser-heated spot was ~20-50 μm across. The heating duration was limited to 3-10 s to avoid temperature fluctuations. It is long enough for all the light elements examined here to diffuse over a ~10 μm size liquid pool[47], which ensures the equilibrium miscible and immiscible liquid states. Previous time-series melting experiments[55] on Fe-S alloys showed that results did not change with increasing heating duration from 1 to 120 s. The one-dimensional temperature profile across a hot spot was obtained by spectro-radiometric method[58]. The temperature at the center of a heated area was used as an upper bound for sample temperature. A lower bound was determined from the one-dimensional temperature profile considering the width of the widest part of a molten area[59]. Pressure was determined from the unit-cell volume of $Al_2O_3$[60] or the Raman shift of diamond after quenching temperature[61]. The contribution of thermal pressure was added by assuming a 5% increase in pressure with each 1000 K increase in temperature, which was found to be consistent with an estimate considering isochoric heating[57].

**XRD measurements and hydrogen concentration in quenched liquid**. The present melting experiments (runs #1-12) were carried in combination with XRD measurements at the beamline BL10XU, SPring-8 synchrotron facility (Supplementary Fig. 3). Angle-dispersive XRD patterns were collected in-situ at high *P-T* by a flat panel detector (Perkin Elmer) using a 6 μm X-ray beam (full width at half maximum) with an energy of ~30 keV. Exposure time was 0.2 s during laser heating and 1 s after quenching temperature. Obtained 2-D XRD images were integrated to conventional 1-D profiles using the IPAnalyzer software[62].

The XRD profiles demonstrate that fcc-, hcp-, or dhcp-$FeH_x$ was formed from H-rich liquid alloys upon quenching temperature. Since hydrogen escapes from iron lattice when its structure changes into bcc during decompression[23], the

hydrogen content $x$ in $FeH_x$ was determined from its volume considering volume expansion due to the incorporation of hydrogen (Supplementary Table 1):

$$x = \frac{V_{sample} - V_{Fe}}{\Delta V_H} \tag{1}$$

in which $V_{Fe}$ is the volume of Fe (ref. [63]) and $\Delta V_H$ is the volume increase of Fe per H atom at the same pressure. $\Delta V_H$ is derived from the volume difference between Fe and FeH calculated by Caracas[64]. In the case of liquid Fe-S-H, FeS crystallized together with $FeH_x$ (Supplementary Fig. 3). FeS exhibited the MnP-type structure (FeS VI) except for in run #1 performed at relatively low pressure of 20 GPa in which the NiAs-type phase (FeS V) was observed[65]. Their volumes for $Z = 4$ are larger by 2-4 Å$^3$ than those calculated from the equation of state of FeS VI[66], which is attributed to the incorporation of hydrogen into FeS crystals. Hydrogen concentration in $FeSH_x$ is calculated to be 0.15-0.25 by using $V_{FeS}$ (Ono et al.[66]) instead of $V_{Fe}$ in Eq. 1, which is consistent with limited volume expansion of FeS by hydrogen incorporation as reported by a previous multi-anvil study[21]. Hydrogen concentrations in carbon-bearing liquids were obtained assuming they were quenched into a mixture of $FeH_x$ and $Fe_7C_3$, whose proportions were estimated by the carbon contents based on EPMA analyses (see below).

These analyses show H/Fe = 0.7-1.1 (molar basis) for homogeneous liquids (Supplementary Table 1), indicating that the variations in hydrogen concentration in liquid were limited in the present experiments as long as liquid phase separation did not occur. When two immiscible liquids were present, the H/Fe molar ratio was found to be 1.1-1.3 in H-rich liquid alloys unless they were enriched in carbon. We assumed that the relative proportions of bubbles and cracks in neighboring H-rich and S-rich (H-poor) liquids represent the difference in the hydrogen content between them, following ref. [22]. It gives the amounts of hydrogen in S-rich liquids to be 10-20% and 70-80% of those in coexisting H-rich liquids in Si/O-free and -bearing samples, respectively.

**Textural and compositional characterizations**. We performed textural and chemical characterizations on recovered samples. A cross section of a laser-heated portion was prepared parallel to the compression axis by a focused Ga ion beam (FIB, FEI, Versa 3D DualBeam). Melting texture and elemental distribution were examined using a field-emission-type scanning electron microscope (FE-SEM) and energy dispersive X-ray spectrometry (EDS). Quantitative chemical analyses of Fe, S, Si, O, and C were then performed by a field-emission-type electron probe microanalyzer (FE-EPMA, JEOL JXA-8530F) with a voltage of 12 kV and a beam current of 15 nA. No coating material was necessary. We used LIF (Fe), PET (S), TAP (Si), LDE1 (O), and LDE2H (C) as analyzing crystals, and Fe, pyrite, silicon, corundum, and $Fe_3C$ as standards. The X-ray counting time for peak/background was 20 s/10 s. The ZAF correction was applied. An almost constant level (0.25-0.37 wt%) of C was found in both a rhenium gasket and a copper grid, which were polished with the FIB together with a sample. Since the gasket and the grid

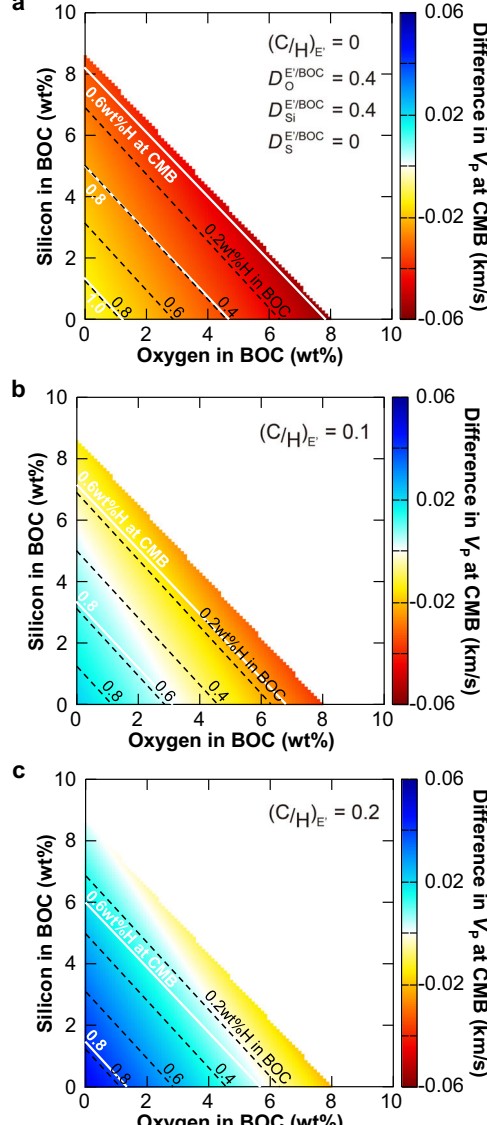

**Fig. 4 Difference in *P*-wave velocity (*V*$_P$) between S-rich bulk outer core (BOC) and H-rich E′ layer.** Colored areas show compositional ranges that explain the density and velocity of the BOC with Si, O, H, and 2 wt% S. The difference in *V*$_P$ of the E′ layer from the BOC at the core-mantle boundary (CMB), and the hydrogen contents in the BOC (black dashed line) and at CMB (white solid line) are given as functions of Si and O in the BOC. These calculations assume the C/H molar ratio in the E′ layer to be 0 in **a**, 0.1 in **b**, and 0.2 in **c**. The dark orange color indicates the *V*$_P$ difference by −0.035 km/s that is supported by seismological models[41–43].

were originally free of carbon, the C detected on them is attributed to contamination during EPMA analyses. That amount of carbon was therefore subtracted from raw analyses.

Quenched metal liquid(s) was found at the center of a laser-heated spot, surrounded either by a solid alloy (liquidus phase), diamond (or hydrocarbon), or Al$_2$O$_3$ pressure medium. Melting of a sample was confirmed by its spherical shape (Fig. 1b) or infiltration of Al$_2$O$_3$ grains into metals (Fig. 1e).

**Compositions of the bulk outer core and the E′ layer**. If the stratification in the uppermost outer core is caused by the liquid immiscibility (Fig. 3), it constrains the chemical compositions of the BOC and the overlying E′ layer. We refer to ref. [18] for the density and sound velocity of liquid iron alloyed with S, H, C, Si, and O. Their calculations along the relatively low, isentropic temperature profile with $T_{ICB}$ = 4800 K at the inner core boundary were adopted because hydrogen is known to reduce the melting temperature of iron substantially[57,67].

The BOC may include 2.0 wt% S (Dreibus and Palme[51]) and a negligible amount of C (see below). The chemical composition of the BOC must be consistent with seismological observations (PREM)[52]. Previous calculations[18] argued that S, H, Si, and O can each be a single light element in the core, as each of them can account for the observed density and compressional-wave velocity over the entire outer core within uncertainties in observations and calculations. With 2.0 wt% S, the remaining hydrogen, silicon and oxygen concentrations are a trade-off (Fig. 4a–c), but they meet:

$$\sum \frac{x_i}{x_i^{max}} = 1 \qquad (2)$$

where $x_i$ is the molar fraction of light element $i$ and $x_i^{max}$ denotes its maximum abundance (23.1 mol% S, 37.8 mol% H, 18.4 mol% Si, and 26.6 mol% O)[18].

The composition of the E′ layer, specifically for its topmost part, is estimated from the possible composition of the BOC. We do not consider sulfur in the E′ layer because H-rich immiscible liquids included the least sulfur in the present experiments (Table 1). Also, run #10 performed at 118 GPa showed that the Si content in H-rich immiscible liquid is about 40% of that in the bulk sample. We apply such relative depletion of silicon for the top of the E′ layer and assume the same for oxygen. Alternatively the E′ layer is relatively enriched in hydrogen and possibly carbon. Depending on the C/H molar ratio (0, 0.1, and 0.2), hydrogen concentration is calculated for the density atop the E′ layer to be the same as (or 0.1 g/cm$^3$ lighter than) that of the BOC when compared at the CMB conditions (Fig. 4a–c and Supplementary Fig. 2). The low velocity of the E′ layer relative to the PREM, −0.035 km/s at its top[41–43], constrains the possible chemical compositions of the E′ layer and consequently the BOC.

## Data availability
The data supporting the main findings of this study are available in the paper and its Supplementary Information. Any additional data can be available from the corresponding authors upon request.

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

## Acknowledgements

We thank N. Hirao and S. Kawaguchi for their assistance in experiments at BL10XU, SPring-8 (proposal no. 2019A0072 and 2019B0072). We also thank K. Yonemitsu for help in sample analyses with FIB and EPMA. This work was supported by JSPS Kakenhi to K.H.

## Author contributions

S.Y. designed and led the project together with K.H., S.Y., S.T., G.M., and Y.O. were involved in DAC experiments and contributed to the manuscript.

## Competing interests

The authors declare no competing interests.
