## [Peer Review File · Nature Communications]

REVIEWER COMMENTS

Reviewer #1 (Remarks to the Author):

General evaluation

This is a well-performed and very interesting experimental contribution, documenting liquid immiscibility in metal alloy compositions relevant to terrestrial planetary cores. The observed liquid immiscibility between S-rich and H-rich cores (from starting compositions also involving C, and in three experiments either Si or O as additional elements) represents new and timely information that give important insights into planetary core evolution. The manuscript is relatively short and very well written and presented. The contribution is therefore welcome, and clearly worth publishing in the scientific literature.

The light element composition of the terrestrial planetary cores are still poorly constrained. Physical core properties, especially for the Earth, where the density and bulk sound velocity structure is fairly well known, provide important information. The iron meteorites demonstrate low-pressure liquid immiscibility between Fe-Ni-dominated melts (with some P) and Fe-Ni-sulphides during their solidification. The heliocentric zoning of oxygen fugacity during core segregation of the terrestrial planets, from very low for the presumed Si-rich Mercury core to more oxidised and very oxidised for the S-rich cores of Mars, Vesta and the iron meteorite parent bodies, might be a guiding feature.

Yokoo et al. use their experimental results to suggest specific scenarios for the evolution and compositional structure of the cores of Mars and Earth. The chosen scenarios, however, appear to be largely ad hoc, without a clear and comprehensive rationale based on planetary materials chemistry, phase relations and physical properties of the Earth, in particular. The article might therefore not have sufficient general interest for Nature Communications (NC). Because NC has less space limitations than e.g. Nature and Nature Geoscience, there might be room for considerable expansion, involving a more detailed and comprehensive introduction to, and discussion of, core segregation and evolution scenarios, putting the liquid immiscibility results for the limited compositional ranges into a broader and more general context. Publication in other specialist journals would clearly also be an option.

Specific comments and suggestions

The inferences made about the BOC and E'-layer compositions and physical properties seem problematic. To be stable over some time period, the E'-layer must have lower density than the BOC.

The Fig. 4 caption is not completely self-explanatory and lacks a reference to the V_p-difference at the top of the E'-layer, relative to the BOC. The references to Helffrich & Kaneshima (2010, Nature) and Kaneshima (2018, PEPI) in the text, might lead the reader to assume that the V_p difference is 0.035 based on the most recent KHOMC model, first introduced by Kaneshima & Helffrich (2013, GJI) and shown e.g. in Kaneshima (2018). It seems that only Fig. 4a or 4b, providing the appropriately dark orange colour, would be applicable. With a C/H-ratio of 0.1 (Fig. 4b), it seems that one needs H-contents of slightly less than 0.6 and 0.2 wt% at the CMB and in the BOC, respectively, combined with about 8 wt% O and almost no Si in the BOC. Combined with 2 wt% S in the BOC, that seems to result in too low density and too high velocity of BOC compared to PREM, using the Fe-alloy property data from Brodholt & Badro (2017, GRL) and Umemoto & Hirose (2020, EPSL).

The restrictions on Si and O in the BOC seem less stringent with a C/H-ratio of 0 (Fig 4a) and H-contents of 0.6-0.7 wt% at the CMB and 0.2-0.3 wt% in BOC. The resulting density and velocity, however, appear be too low and too high, respectively, also in this case. The S (2 wt%) and H (0.2-0.3 wt%) contents contribute to the deviations from PREM.

Based on Fig. 3 of Umemoto & Hirose (2020), H has an extreme leverage on the physical properties of the outer core, and it seems impossible to explain reduced seismic velocity of an E'-layer mainly by slightly increased H content in such a layer. The extreme leverage is seen after recalculating and plotting the Umemoto & Hirose (2020) Fe-H curve with the corresponding curves for Fe-S, Fe-Si, Fe-O and Fe-C of Brodholt & Badro (2017) and Umemoto & Hirose with a weight% abscissa axis. My calculations show

that introducing 0.2 wt% H (from 0% H) will increase the velocity by about 58 m/s. The net effect of increasing the H-content in an E'-layer of the suggested magnitude, may be to increase the velocity to such an extent that it seems unrealistic to be able to compensate for it by reducing mainly S, and possibly some Si and O, as suggested, matching the required density **and** velocity reductions. Increasing the H-content from 0.2 wt% (assumed for BOC; Yokoo et al.'s suggestion: <0.4) to 0.6 wt% (assumed for top of E'; Yokoo et al.'s suggestion: 0.4-0.8), will increase the velocity by 110 m/s and reduce the density by 510 kg/m³. With a maximum V_p-reduction of about 36 m/s in the outermost core (just below CMB), based on the KHOMC-model of Kaneshima & Helffrich (2013) and Kaneshima (2018), combined with a density reduction of approximately 100 kg/m³ (about 1 %), in general accordance Brodholt and Badro (2017), it seems difficult to match the suggested H-concentration variations with the mineral physics effects of changing the concentrations of the other light element candidates.

In terms of establishing a context for core formation and evolution of the Earth, the equilibrium:

which is displaced to the right with increasing temperatures (relevant to high-T core segregation) and reversed during planetary cooling, seems to be an essential component (e.g. Takafuji et al. 2005, GRL; Frost et al. 2010; Tsuno et al. 2013, GRL; Hernlund & Geissman, 2016, SEDI-abstr; Hirose et al. 2017, Nature; Laneuville et al. 2018, PEPI; Trønnes et al. 2019, Tectonophysics; Helffrich et al. 2020, GRL). The exact nature and physical properties of the E'-layer are still very uncertain. The velocity structure of Kaneshima (2018) and the mineral physics properties of Badro et al. (2015, PNAS) and Brodholt and Badro (2017), however, seem consistent with cooling-associated loss of SiO₂ (via liquid-state diffusion or floating crystals) to the initial magma ocean (MO) and the subsequent basal magma ocean (BMO), coupled with the incorporation of FeO (more than twice as many mol-units as SiO₂ are required) from the MO and BMO (e.g. Hernlund & Geissman, 2016; Brodholt & Badro, 2017; Trønnes et al. 2019). Such protocore-MO and core-BMO exchanges of SiO₂ and FeO appear inescapable for the Earth, and probably for Venus also. This **does not exclude** minor amounts of other light elements like H, S and C in the BOC and E', but specific core compositions should preferably be discussed in as wide a context as possible.

It is interesting to note that Li et al. (2020, GRL) states that: "A core with 1 wt% hydrogen is equivalent to the hydrogen content of ~130 oceans of water. Even with our highest partition coefficients at 50 GPa and 3,500 K, a core with that much water would result in a mantle content of some 23 oceans of water. With one ocean at the surface now, this implies that the current mantle should have 22 oceans of water, which is above even the most optimistic saturation limits for the mantle²⁵, unless bridgmanite is able to incorporate far more water than is currently expected²⁶." With only 0.1wt% H in the core, the corresponding mantle water content would likely be significantly lower, and possibly at an acceptable level, in terms of the bridgmanite carrying capacity.

Minor suggestions

As pointed out above, the manuscript is well written and presented, with well-designed and appropriate figures and tables. I have made a few specific suggestions.

Line 32: I suggest: -- between liquids **with** Fe-S(±Si/O) and Fe-H(±C), based on ---

L45 - L87: General comment on verb tenses: When you describe your specific operations, related to the experimental work, a simple past tense is appropriate, just as you have used in L45-L48 (first half of L48). However, when you describe observations, e.g. based on your run products (which you still have, presumably), it is better to use the present tense. Below, I have identified specific text segment in the L45-L87 section where the verb tense could be changed. You might be able to identify other sentences in other parts of the manuscript, where I have not suggested corrections(?):

L48 (second half)-L48: When liquids **are** relatively enriched in carbon, the textures of the two immiscible liquids **are** very similar --- (of course you cannot observe the liquids, which are quenched (and you don't need to say *quenched* liquids), but this is a general observation or general conclusion based on your observations: therefore present tense.

Further on:

L52 - L56: --- liquids **coexist** ---, --- liquid **exhibits** more bubbles --- (or, better: --- liquid **contains** more bubbles ---). But **it should be**: --- **escaped** during decompression ---, --- liquids **were** found to be --- and --- Some experiments **showed** a change ---.

L57 - L58: --- melt pools **are** present, we found a homogeneous Fe-S-H liquid in ---

- L60 - L62: ---, the sulfur concentration in the liquid gradually **decreases** toward the outside and sharply **increases** at the edge. ---
- L64 - L66: ---their bulk composition **is** identical to that of a homogeneous portion at the center of a heated area (runs #1 & 2, Fig. 1f). Such microscopic separation **is** likely caused by --
- L69-L70: --- The experiments on Si- and O-bearing samples **demonstrates** that when immiscible S-rich and H-rich liquids **coexist**, both silicon and oxygen **are** preferentially ---
- L71-L74: ---, carbon **is** included more in the latter above 71 GPa, while it **is** partitioned almost equally at lower pressures (Table 1). These **relations** suggest that liquid Fe alloys **can be** classified into two groups, ----
- L81-L87: --- **The boundary has** a relatively steep increase in T below ~60 GPa. The two coexisting immiscible liquids **contain** 13–19 wt% S + 0.2–0.4 wt% H and 0–1.5 wt% S + 0.7–1.9 wt% H, respectively, indicating a wide miscibility gap between S-rich and H-rich liquids (Table 1). On the other hand, when silicon or oxygen **are** involved, the sulfur concentrations in the immiscible S-rich and H-rich liquids **are** 2–8 wt% and 0–0.3 wt%, respectively, indicating a smaller gap in sulfur abundance in the **presence of silicon and/or oxygen**.
- L269: --- The diamond anvils had flat or bevelled culets with diameters of 300 and 120 μm , respectively. --
- (Note: a bevel, but a bevelled surface. You cannot say: flat or bevelled culet size)
- L295: --- were carried **in combination with** X-ray diffraction (XRD) ---
- L466: Please specify the error values, e.g. **1 σ** or **2 σ** : --- indicate errors (**1 σ**) in the last digits.
or: --- indicate errors (**2 σ**) in the last digits.

Extended Data Table 1, Starting Materials:

It looks like Al_2O_3 is part of the starting material. Please replace e.g. Fe10S+P+A and the footnote: A = Al_2O_3 by:

Fe10S+P, A and the footnote: **A = pressure medium, Al_2O_3**

Reviewer #2 (Remarks to the Author):

The authors performed new iron-alloy melting experiments in diamond-anvil cell at High-Pressures (HP) (between 20 and 118 GPa) and High-Temperatures (HT) (between 1990 and 3620 K). They reported the immiscibility of Fe-S (+ Si,O) liquids and Fe-H (+ C) liquids above 20 GPa. It turns out that 20 GPa is also a relevant pressure for the core of Mars. This first result is therefore very important for the history of this planet with a possible explanation for the existence and then the extinction of its magnetic field. Then, the authors also observed an immiscibility of Fe-S-H (+Si, O, C) around 118 GPa suggesting that this phenomenon could occur in the earth's core. With such a possible immiscibility at the topmost outer core, this study would explain the formation of a low-velocity layer at the top of the core (named E') observed by seismology. These results may represent a very important scientific breakthrough in the knowledge of planetary cores. However, I have several important concerns about this study, which are detailed below.

1. Generally speaking, the way of writing or explaining is not always clear, making the manuscript sometimes difficult to understand. Short and concise sentences would make it easier to understand. Passages that should be written in a simpler way are: line 57 ("When a few different, ..."), lines 86-87, lines 98-99 and lines 143-151 (sentences too long and unclear).
2. An extremely annoying point in these experiments is that no chemical element mass balance was performed after loading the diamond cell (before the HP-HT runs). We know that the authors added C_nH_{2n+2} paraffin and Al_2O_3 in some experiments but we don't know their relative proportions of each phases. However, the chemical characterization of the starting materials (just before the HP-HT experiments) remains essential here to know if the contents calculated by the authors after the HP-HT experiments are consistent with their starting systems. A mass balance (before and after the HP-HT reactions) would allow to fully validate the numerous hypotheses made for the calculations of the H contents in particular... For example, in lines 303-306, it is written that: "Since hydrogen escapes from iron lattice when its structure changes into bcc during decompression, the hydrogen content x in FeH_x was determined from its volume considering volume expansion due to the incorporation of hydrogen ": on which precise results is this remark based (about the decompression and the volumes used)? What are the errors that lead to these hypotheses on the calculations presented in the rest of this study? Furthermore, the errors indicated in Table 1 seem to be very small with respect to the method and assumptions made to obtain these results. A calculation explaining and justifying these errors is missing here. Indeed, the errors presented in Table 1 for the contents (calculated and not directly measured) of H can fall below 9%, which is better to errors found on measurements made directly by nuclear microprobe for example. Lines 340-344: The origin of these C contents is not clear. The authors should better argue with a SEM photo or better explain these observations that they consider as artifacts? This is important because significant amounts of C are subtracted from their raw analyses. Lines 52-54 and lines 326-329: How did the authors calculate H concentrations from a volume (of bubbles or cracks just from 2D photos?)? What methods did they use? What values did they take for the densities involved? What are the errors on the H concentrations found by this method? Are the percentages written (10-20% and 70-80%) wt%, mol% or vol%?
3. Lines 114-117: What would be the consequences with regard to the silicates present in the magma ocean? Would these consequences be different if the Fe-S-H immiscibility appears once the core is formed and not during its segregation?
4. Lines 282: "The heating was limited to 3-10 sec...". This heating time is extremely short. It would be important to justify that it is enough for each element (H, Si, C or S) by a calculation based on the two references mentioned for example.
5. There are obviously significant temperature gradients throughout the experiments that could have prevented (at least chemical) equilibrium. In other words, can the calculated contents of H, C, S, and Si obtained in this study be used quantitatively as a starting point for modeling for the Earth and Martian core?
6. Extended Data Fig 2 b and c : I do not see the border between the two "liquids" where the authors drew it with the dotted lines... On picture 2b: I see on the left and right edges a contrast identical to the one above. On photo 2c: I see no difference in the contrasts (maybe it is due to the quality of my pdf file).
7. Lines 347: I don't see the spherical shape or infiltration of Al_2O_3 grains into metal (photo 1e).

Reviewer #3 (Remarks to the Author):

The manuscript report experimental results on the immiscibility of the Fe-S-H system to Mbar pressures. The results were used for the discussions on the formation and termination of the Martian magnetic field, as well as the nature of the low-velocity layer below the core-mantle boundary in Earth's outer core. The experiments are certainly very challenging, especially for experiments approaching the Earth's core conditions. That is the reason why there are so few data points available. I find the implications of the data intriguing, but the conclusions are based on the sparse data. In particular, the discussion on the shallow part of the Earth's outer core (or E' layer) is speculative. Therefore, I do not recommend the publication of the manuscript in its present form.

Main issues:

- The miscible/immiscible boundary was fitted from the very limited data. There supposed to be huge uncertainties in the fitted parameters (Line 78-81). In Fig. 2, dashed lines below and above the immiscible boundary (black line) were not described. I suppose these are confidence bands of the fittings.
- The constraints for the boundary are mainly from the three data points data below 40 GPa and the higher pressures data points only provide the lower bound. It is questionable how one can reliably fit the data to constrain the miscible/immiscible boundary up to the earth's core pressures. It will make a convincing case, if the authors can provide a few more data points at > 40-120 GPa where the Fe-S-H liquids is miscible.

Minor comments:

Line 48: It is better to specify the carbon content for the liquids.

Line 54-55: It is quite vague to state "relatively high temperatures" here. What temperature is considered high temperature?

Line 338-340: Please describe in details the analysis of carbon in the Fe alloy samples, as we know it can be tricky and challenging to measure carbon by EPMA.

Table 1: How was the hydrogen content of immiscible liquids determined for run #3, 6, 7-11?

Fig. 1d: the texture analysis of run #6 needs more description. Why does it appear that there are two domains in the miscible liquid: one is enriched in H and the other depleted in H? What are those black domains between the two liquid pockets?

Response to comments by Reviewer #1 (reviewer comments in bold and our response in blue):

This is a well-performed and very interesting experimental contribution, documenting liquid immiscibility in metal alloy compositions relevant to terrestrial planetary cores. The observed liquid immiscibility between S-rich and H-rich cores (from starting compositions also involving C, and in three experiments either Si or O as additional elements) represents new and timely information that give important insights into planetary core evolution. The manuscript is relatively short and very well written and presented. The contribution is therefore welcome, and clearly worth publishing in the scientific literature.

The light element composition of the terrestrial planetary cores are still poorly constrained. Physical core properties, especially for the Earth, where the density and bulk sound velocity structure is fairly well known, provide important information. The iron meteorites demonstrate low-pressure liquid immiscibility between Fe-Ni-dominated melts (with some P) and Fe-Ni-sulphides during their solidification. The heliocentric zoning of oxygen fugacity during core segregation of the terrestrial planets, from very low for the presumed Si-rich Mercury core to more oxidised and very oxidised for the S-rich cores of Mars, Vesta and the iron meteorite parent bodies, might be a guiding feature.

Yokoo et al. use their experimental results to suggest specific scenarios for the evolution and compositional structure of the cores of Mars and Earth. The chosen scenarios, however, appear to be largely ad hoc, without a clear and comprehensive rationale based on planetary materials chemistry, phase relations and physical properties of the Earth, in particular. The article might therefore not have sufficient general interest for Nature Communications (NC). Because NC has less space limitations than e.g. Nature and Nature Geoscience, there might be room for considerable expansion, involving a more detailed and comprehensive introduction to, and discussion of, core segregation and evolution scenarios, putting the liquid immiscibility results for the limited compositional ranges into a broader and more general context. Publication in other specialist journals would clearly also be an option.

In response to the Reviewer's comments, we have augmented the Introduction by mentioning the consequences of liquid immiscibility in planetary cores as follows;

“Metallic iron cores of terrestrial planets and moons likely contain more than one light elements^{1,2}. Multiple light elements in molten iron often cause liquid-liquid immiscibility at 1 bar and high pressure such as in Fe-O, Fe-S-O, Fe-S-Si and Fe-S-C³⁻⁹. The liquid immiscibility in planet/satellite iron cores gives rise to their stable stratification. At the same time, the density difference between immiscible two liquids can drive core convection, leading to a planetary

dynamo action and magnetic field generation for a period of time.”

We have also added statements in a broader context in the Discussion part. See detailed response below.

Specific comments and suggestions

The inferences made about the BOC and E'-layer compositions and physical properties seem problematic. To be stable over some time period, the E'-layer must have lower density than the BOC.

Certainly it is straightforward to assume a lower density in the E' layer for its stability. However, a recent seismological study showed that the outermost core exhibits a higher density than that of PREM (the reference core density) (van Tent et al. 2020, GJI). Considering uncertainties in such seismological observations, it is reasonable to assume “neutral” density in this paper.

Nevertheless, by following this Reviewer’s comment, we have added a new figure as Supplementary Fig. 2 (see below), which demonstrates the case with 0.1 g/cm^3 smaller density in the E' layer than that of the BOC when compared at CMB conditions. In this case, the V_P difference of -0.035 km/s is explained by relatively O-rich ($>7 \text{ wt\% O}$) composition in the E' layer when C is absent.

The relevant, following statements were also added in the Discussion (Lines 167–169);

“On the other hand, with 0.1 g/cm^3 smaller density in the E' layer than in the BOC when compared at the CMB, the H contents are $<0.2 \text{ wt\%}$ in the BOC and $\sim 0.7 \text{ wt\%}$ at the CMB (Supplementary Fig. 2).”

Supplementary Fig. 2 Difference in V_P between S-rich bulk outer core (BOC) and H-rich E' layer with a smaller density. Conditions are the same as those for Fig. 4a except that the E' layer is lighter by 0.1 g/cm^3 than that of the BOC when compared at the CMB.

The Fig. 4 caption is not completely self-explanatory and lacks a reference to the V_p -difference at the top of the E'-layer, relative to the BOC. The references to Helffrich & Kaneshima (2010, Nature) and Kaneshima (2018, PEPI) in the text, might lead the reader to assume that the V_p difference is 0.035 based on the most recent KHOMC model, first introduced by Kaneshima & Helffrich (2013, GJI) and shown e.g. in Kaneshima (2018).

Following this comment, we have added a statement at the end of the Fig. 4 caption;

“The dark orange colour indicates the V_p difference by -0.035 km/s that is supported by seismological models⁴¹⁻⁴³.”

Also, Kaneshima & Helffrich (2013, GJI) was added as a reference in the main text.

It seems that only Fig. 4a or 4b, providing the appropriately dark orange colour, would be applicable. With a C/H-ratio of 0.1 (Fig. 4b), it seems that one needs H-contents of slightly less than 0.6 and 0.2 wt% at the CMB and in the BOC, respectively, combined with about 8 wt% O and almost no Si in the BOC. Combined with 2 wt% S in the BOC, that seems to result in too low density and too high velocity of BOC compared to PREM, using the Fe-alloy property data from Brodholt & Badro (2017, GRL) and Umemoto & Hirose (2020, EPSL).

This is a confusion because of a complex organization of Umemoto and Hirose (2020, EPSL). If one refers to Fig. 6 in Umemoto and Hirose (2020) which considers uncertainties in both observations and calculations, the BOC compositions that we found acceptable (e.g. Fe + 4.5 wt% O + 0.4 wt% H + 2 wt% S, no Si and C) are compatible with the outer core density and velocity.

Figure 6 in Umemoto & Hirose (2020, EPSL) ($T_{ICB} = 4800$ K)

The restrictions on Si and O in the BOC seem less stringent with a C/H-ratio of 0 (Fig 4a) and H-contents of 0.6-0.7 wt% at the CMB and 0.2-0.3 wt% in BOC. The resulting density and velocity, however, appear to be too low and too high, respectively, also in this case. The S (2 wt%) and H (0.2-0.3 wt%) contents contribute to the deviations from PREM. Based on Fig. 3 of Umemoto & Hirose (2020), H has an extreme leverage on the physical

properties of the outer core, and it seems impossible to explain reduced seismic velocity of an E'-layer mainly by slightly increased H content in such a layer. The extreme leverage is seen after recalculating and plotting the Umemoto & Hirose (2020) Fe-H curve with the corresponding curves for Fe-S, Fe-Si, Fe-O and Fe-C of Brodholt & Badro (2017) and Umemoto & Hirose with a weight% abscissa axis. My calculations show that introducing 0.2 wt% H (from 0% H) will increase the velocity by about 58 m/s. The net effect of increasing the H-content in an E'-layer of the suggested magnitude, may be to increase the velocity to such an extent that it seems unrealistic to be able to compensate for it by reducing mainly S, and possibly some Si and O, as suggested, matching the required density and velocity reductions. Increasing the H-content from 0.2 wt% (assumed for BOC; Yokoo et al.'s suggestion: <0.4) to 0.6 wt% (assumed for top of E'; Yokoo et al.'s suggestion: 0.4-0.8), will increase the velocity by 110 m/s and reduce the density by 510 kg/m³. With a maximum V_p-reduction of about 36 m/s in the outermost core (just below CMB), based on the KHOMC-model of Kaneshima & Helffrich (2013) and Kaneshima (2018), combined with a density reduction of approximately 100 kg/m³ (about 1 %), in general accordance Brodholt and Badro (2017), it seems difficult to match the suggested H-concentration variations with the mineral physics effects of changing the concentrations of the other light element candidates.

The Reviewer is concerned that increasing the H content changes the density and velocity too much, but it should not be a problem because the amount of H we discuss here is smaller than those of other light elements when expressed in wt%.

For clarity, we summarize below the effect of each light element on the density and velocity of liquid Fe at CMB conditions from Fig. 3 in Umemoto & Hirose (2020). These are slightly smaller than but similar to the values used by the Reviewer's argument.

Table R1. The effect of each light element on the density and velocity of liquid Fe in at% and wt%.

	$\Delta\rho$ (kg/m ³) per 1at%	ΔV_P (m/s) per 1at%	$\Delta\rho$ (kg/m ³) per 1wt%	ΔV_P (m/s) per 1wt%	Increase in V_P (m/s) per ρ reduction of 1kg/m ³
H	-31	7	-1104	233	0.21
C	-33	17	-150	74	0.50
O	-41	12	-140	41	0.29
Si	-59	15	-117	29	0.25
S	-46	14	-80	25	0.31

The effect of H is certainly larger than those of others when adding each light element by 1 wt%. However, the difference in the H content (wt%) between the E' layer and the BOC is

much smaller than the 1 wt%. For example, the effect of 0.4 wt% H on density reduction ($\sim 440 \text{ kg/m}^3$) is equivalent to the combined effects of 2 wt% S + ~ 2 wt% O or 2 wt% S + ~ 2.4 wt% Si. It also has an effect of increasing V_P ($\sim 92 \text{ m/s}$), which is indeed smaller than those by 2 wt% S + 2 wt% O ($50 + 82 \text{ m/s}$) or 2 wt% S + 2.4 wt% Si ($50 + 70 \text{ m/s}$). It is shown in Table R1 (see the right most column) that the effect of H on velocity with respect to that on density is the smallest among these light elements. Thus, replacing H for other elements results in lower velocity when density does not change. This suggests that the E' layer may be enriched in H as we propose in this paper.

In response to this comment, we have added the following statements on why hydrogen is enriched in the E' layer in Line 156–159;

“...the E' layer could be depleted in S, Si and O but instead enriched in H and C compared to the BOC. Indeed, such compositional difference causes the low-velocity anomaly in the E' layer (Fig. 4), because the effect of hydrogen on increasing liquid Fe velocity with respect to decreasing its density is smaller than those of other light elements and thus replacing hydrogen for other elements without changing density results in lower velocity.”

In terms of establishing a context for core formation and evolution of the Earth, the equilibrium: $2\text{Fe}^{\text{in metal}} + \text{SiO}_2^{\text{in silicate melt}} = \text{Si}^{\text{in metal}} + 2 \text{FeO}^{\text{in silicate melt}}$, which is displaced to the right with increasing temperatures (relevant to high-T core segregation) and reversed during planetary cooling, seems to be an essential component (e.g. Takafuji et al. 2005, GRL; Frost et al. 2010; Tsuno et al. 2013, GRL; Hernlund & Geissman, 2016, SEDI-abstr; Hirose et al. 2017, Nature; Laneuville et al. 2018, PEPI; Trønnes et al. 2019, Tectonophys; Helffrich et al. 2020, GRL). The exact nature and physical properties of the E'-layer are still very uncertain. The velocity structure of Kaneshima (2018) and the mineral physics properties of Badro et al. (2015, PNAS) and Brodholt and Badro (2017), however, seem consistent with cooling-associated loss of SiO₂ (via liquid-state diffusion or floating crystals) to the initial magma ocean (MO) and the subsequent basal magma ocean (BMO), coupled with the incorporation of FeO (more than twice as many mol-units as SiO₂ are required) from the MO and BMO (e.g. Hernlund & Geissman, 2016; Brodholt & Badro, 2017; Trønnes et al. 2019). Such protocore-MO and core-BMO exchanges of SiO₂ and FeO appear inescapable for the Earth, and probably for Venus also. This does not exclude minor amounts of other light elements like H, S and C in the BOC and E', but specific core compositions should preferably be discussed in as wide a context as possible.

As the Reviewer pointed out, Brodholt and Badro (2017 GRL) argued that the E' layer is depleted in Si and enriched in O compared to the bulk outer core (BOC) as a consequence of

core-BMO chemical reaction. However, as Helffrich (2014 EPSL) pointed out, the diffusivities of Si and O are not high enough to form 300-km thick E' layer (note that Broadholt and Badro considered a light liquid for the E' layer and such buoyant liquid at the top is not involved in convection). In addition, it is likely that the formation of ferropericlasite layer separates the BMO from the core and limits chemical reaction between them.

Here we compare the density of the BMO with that of ferropericlasite at CMB conditions. Ferropericlasite starts crystallizing after ~50% solidification of a pyrolitic magma ocean (Caracas et al. 2019, EPSL). **Fig. R1** (see below) shows the density of the BMO after 50% bridgmanite crystallization that was calculated by Caracas and others, in comparison with that of ferropericlasite whose Fe content is estimated from bridgmanite/melt Fe partitioning ($D_{\text{Fe}} = 0.5$) (Andraut et al. 2012, Nature) and the Mg/Fe exchange coefficient (1.0) between bridgmanite and ferropericlasite (Sinmyo & Hirose 2013, PCM). We employ the thermal equations of state of low-spin ferropericlasite reported by Mao et al. (2011, GRL) and Ricolleau et al. (2009, GRL) and the zero-pressure volume V_0 considering the Fe content from Fei et al. (2007, GRL).

Fig. R1. Densities of a magma ocean (blue) after 50% solidification (Caracas et al., 2019) and crystallizing ferropericlasite (red, Mao et al. 2011; orange, Ricolleau et al. 2009) at 4,000 K. Density of MgO is also shown for comparison (green, Ricolleau et al. 2009). While different thermal effects were estimated in the previous studies, the density of ferropericlasite at the CMB is close to that of the magma ocean in either case. The temperature at the 50% solidification of the magma ocean can be higher than 4,000 K, which supports ferropericlasite is denser than the magma ocean.

Fig. R1 demonstrates that the BMO can crystallize denser ferropericlasite after 50% solidification. It depends on its thermal EoS chosen, but the temperature at the 50% solidification is likely to be higher than 4,000 K, which supports the crystallization of denser ferropericlasite. While bridgmanite floats in the BMO, a dense ferropericlasite layer will form between the BMO and the liquid core, which suppresses the chemical interaction between them.

The core-BMO reaction before 50% solidification should be minor because the downward diffusion of oxygen in the liquid core should have been limited. We therefore do not consider the effect of chemical interaction between the BMO and the liquid core.

Based on this comment, we now argue the chemical reaction between the core liquid and the BMO in Line 136–142 as;

“Chemical reaction of core metal with an FeO-rich basal magma ocean (BMO)⁴⁶ results in the depletion in Si and the enrichment in O, which can produce a light and slow layer at the top of the core⁴⁵, but the limited diffusivities of these elements may prohibit the formation of ~300-km thick E’ layer⁴⁷. Furthermore, the BMO may have crystallized bridgmanite that floats above⁴⁸ and then dense ferropericlase^{49,50} that forms a layer separating the BMO and the core and suppresses the chemical reaction between them.”

It is interesting to note that Li et al. (2020, GRL) states that: "A core with 1 wt% hydrogen is equivalent to the hydrogen content of ~130 oceans of water. Even with our highest partition coefficients at 50 GPa and 3,500 K, a core with that much water would result in a mantle content of some 23 oceans of water. With one ocean at the surface now, this implies that the current mantle should have 22 oceans of water, which is above even the most optimistic saturation limits for the mantle²⁵, unless bridgmanite is able to incorporate far more water than is currently expected²⁶." With only 0.1wt% H in the core, the corresponding mantle water content would likely be significantly lower, and possibly at an acceptable level, in terms of the bridgmanite carrying capacity.

The most recent experimental work by Tagawa et al. (2021, Nature Commun.) on the metal-silicate partitioning of hydrogen demonstrated that hydrogen is strongly siderophile at the high *P-T* conditions of core formation. Their core formation models suggest that 0.3–0.6 wt% H was incorporated into the Earth’s core leaving only 700 ppm H₂O in the magma ocean, which corresponds to the sum of water in oceans and in the mantle (Hirschmann 2016, AmMin). The possible range of the BOC composition we propose in Fig. 4 (<0.4 wt% H) overlaps with the results of the core formation modeling by Tagawa et al. (2021). Based on this discussion, we added a sentence in Line 165–167 as follows;

“Such possible range of the BOC composition is consistent with a recent core formation model²⁰ which suggests that 0.3–0.6 wt% H was incorporated into the Earth’s core-forming metals.”

Minor suggestions

As pointed out above, the manuscript is well written and presented, with well-designed and appropriate figures and tables. I have made a few specific suggestions.

Line 32: I suggest: -- between liquids with Fe-S(\pm Si/O) and Fe-H(\pm C), based on ---

L45 - L87: General comment on verb tenses: When you describe your specific operations, related to the experimental work, a simple past tense is appropriate, just as you have used in L45-L48 (first half of L48). However, when you describe observations, e.g. based on your run products (which you still have, presumably), it is better to use the present tense. Below, I have identified specific text segment in the L45-L87 section where the verb tense could be changed. You might be able to identify other sentences in other parts of the manuscript, where I have not suggested corrections(?):

L48 (second half)-L48: When liquids are relatively enriched in carbon, the textures of the two immiscible liquids are very similar --- (of course you cannot observe the liquids, which are quenched (and you don't need to say quenched liquids), but this is a general observation or general conclusion based on your observations: therefore present tense.

Further on:

L52 - L56: --- liquids coexist ---, --- liquid exhibits more bubbles --- (or, better: --- liquid contains more bubbles ---). But it should be: --- escaped during decompression ---, --- liquids were found to be --- and --- Some experiments showed a change ---.

L57 - L58: --- melt pools are present, we found a homogeneous Fe-S-H liquid in ---

L60 - L62: ---, the sulfur concentration in the liquid gradually decreases toward the outside and sharply increases at the edge. ---

L64 - L66: ---their bulk composition is identical to that of a homogeneous portion at the center of a heated area (runs #1 & 2, Fig. 1f). Such microscopic separation is likely caused by --

L69-L70: --- The experiments on Si- and O-bearing samples demonstrates that when immiscible S-rich and H-rich liquids coexist, both silicon and oxygen are preferentially --
-

L71-L74: ---, carbon is included more in the latter above 71 GPa, while it is partitioned almost equally at lower pressures (Table 1). These relations suggest that liquid Fe alloys can be classified into two groups, ----

L81-L87: --- The boundary has a relatively steep increase in T below ~60 GPa. The two coexisting immiscible liquids contain 13–19 wt% S + 0.2–0.4 wt% H and 0–1.5 wt% S + 0.7–1.9 wt% H, respectively, indicating a wide miscibility gap between S-rich and H-rich liquids (Table 1). On the other hand, when silicon or oxygen are involved, the sulfur concentrations in the immiscible S-rich and H-rich liquids are 2–8 wt% and 0–0.3 wt%, respectively, indicating a smaller gap in sulfur abundance in the presence of silicon and/or oxygen.

L269: --- The diamond anvils had flat or bevelled culets with diameters of 300 and 120 μm , respectively. --- (Note: a bevel, but a bevelled surface. You cannot say: flat or bevelled

culet size)

L295: --- were carried in combination with X-ray diffraction (XRD) ---

L466: Please specify the error values, e.g. 1σ or 2σ : --- indicate errors (1σ) in the last digits. or: --- indicate errors (2σ) in the last digits.

Extended Data Table 1, Starting Materials: It looks like Al_2O_3 is part of the starting material. Please replace e.g. $\text{Fe}_{10}\text{S}+\text{P}+\text{A}$ and the footnote: $\text{A} = \text{Al}_2\text{O}_3$ by: $\text{Fe}_{10}\text{S}+\text{P}$, A and the footnote: $\text{A} =$ pressure medium, Al_2O_3

All of these suggestions were incorporated into the revised manuscript.

We thank the Reviewer for his valuable comments, which helped to strengthen our discussion.

Response to comments by Reviewer 2 (reviewer comments in bold and our response in blue):

The authors performed new iron-alloy melting experiments in diamond-anvil cell at High-Pressures (HP) (between 20 and 118 GPa) and High-Temperatures (HT) (between 1990 and 3620 K). They reported the immiscibility of Fe-S (+ Si,O) liquids and Fe-H (+ C) liquids above 20 GPa. It turns out that 20 GPa is also a relevant pressure for the core of Mars. This first result is therefore very important for the history of this planet with a possible explanation for the existence and then the extinction of its magnetic field. Then, the authors also observed an immiscibility of Fe-S-H (+Si, O, C) around 118 GPa suggesting that this phenomenon could occur in the earth's core. With such a possible immiscibility at the topmost outer core, this study would explain the formation of a low-velocity layer at the top of the core (named E') observed by seismology. These results may represent a very important scientific breakthrough in the knowledge of planetary cores.

We thank the Reviewer for his/her encouraging comment.

However, I have several important concerns about this study, which are detailed below.

1. Generally speaking, the way of writing or explaining is not always clear, making the manuscript sometimes difficult to understand. Short and concise sentences would make it easier to understand. Passages that should be written in a simpler way are: line 57 ("When a few different, ..."), lines 86-87, lines 98-99 and lines 143-151 (sentences too long and unclear).

These suggestions are helpful. We rewrote all of these sentences for clarity.

2. An extremely annoying point in these experiments is that no chemical element mass balance was performed after loading the diamond cell (before the HP-HT runs). We know that the authors added $\text{C}_n\text{H}_{2n+2}$ paraffin and Al_2O_3 in some experiments but we don't

know their relative proportions of each phases. However, the chemical characterization of the starting materials (just before the HP-HT experiments) remains essential here to know if the contents calculated by the authors after the HP-HT experiments are consistent with their starting systems.

In the present experiments, Al_2O_3 was not involved in chemical reaction because the solubility of Al in liquid Fe is negligible at the present experimental temperature range. As shown in Fig. R2 below, $\text{C}_n\text{H}_{2n+2}$ paraffin worked as a hydrogen source, leaving diamond and hydrocarbon compound after high P - T experiments. The C/H ratio of such hydrocarbon (green region in the left-hand panel) is not known, making mass-balance calculations difficult. Nevertheless, the miscibility/immiscibility of liquid Fe-S-H observed in our experiments should represent the equilibrium liquid states.

Fig R2. X-ray elemental maps for C (left) and Fe + S (right) (Fig. 1b) around a quenched liquid pool obtained in run #7. Both diamond (red) and hydrocarbon (green) were found in the C map.

A mass balance (before and after the HP-HT reactions) would allow to fully validate the numerous hypotheses made for the calculations of the H contents in particular... For example, in lines 303-306, it is written that: "Since hydrogen escapes from iron lattice when its structure changes into bcc during decompression, the hydrogen content x in FeH_x was determined from its volume considering volume expansion due to the incorporation of hydrogen": on which precise results is this remark based (about the decompression and the volumes used)?

We primarily base this statement on recent neutron diffraction experiments on Fe-H alloys, in which both hydrogen concentrations and unit-cell volumes were determined (Machida et al. 2014, Nature Commun.; Iizuka-Oku et al. 2017, Nature Commun.; Ikuta et al. 2019, Sci. Rep.). In this study as well as in Tagawa et al. (2021 Nature Commun.), we obtained the hydrogen contents in Fe alloys from their unit-cell volumes obtained by synchrotron XRD measurements.

In response to this comment, we added a reference (Iizuka-Oku et al. 2017) in this sentence.

What are the errors that lead to these hypotheses on the calculations presented in the rest of this study? Furthermore, the errors indicated in Table 1 seem to be very small with respect to the method and assumptions made to obtain these results. A calculation explaining and justifying these errors is missing here. Indeed, the errors presented in Table 1 for the contents (calculated and not directly measured) of H can fall below 9%, which is better to errors found on measurements made directly by nuclear microprobe for example.

The H content in Fe alloy has been determined with Eq. 1. In the original manuscript, as a source of its error, we considered only uncertainty in the volume expansion, namely errors in pressure and the volume of iron lattice. Thanks to this Reviewer's comment, we now additionally consider the uncertainty in ΔV_H , the volume increase per hydrogen atom, based on the neutron diffraction study by Ikuta et al. (2019). The data only obtained for a single-phase fcc by Ikuta et al. (2019) yields $\pm 8\%$ error in ΔV_H . Accordingly, we revised the errors in the H contents in Table 1. They are now 9–20%.

Lines 340-344: The origin of these C contents is not clear. The authors should better argue with a SEM photo or better explain these observations that they consider as artifacts? This is important because significant amounts of C are subtracted from their raw analyses.

We subtracted 0.3–0.4 wt% C from raw EPMA analyses, which is not significant but is sometimes important as the Reviewer pointed out.

In the present DAC experiments, sample was loaded into a rhenium gasket. After high P - T experiments, we prepared a cross section of the sample together with the gasket by using an FIB. A thin slice of the cross section was attached to a copper grid, and subsequently the surface of the sample and the gasket were polished with the FIB before EPMA analyses (the copper grid was also processed together). During the EPMA analyses, we always find an almost constant level (0.25–0.37 wt%) of C in both the rhenium gasket and the copper grid. Since both are originally free of carbon, the detection of C should be contamination during EPMA analyses.

In response to this comment, we have augmented the relevant statement in Lines 254–257 as follows;

“An almost constant level (0.25–0.37 wt%) of C was found in both a rhenium gasket and a copper grid, which were polished with the FIB together with a sample. Since the gasket and the grid were originally free of carbon, the C detected on them is attributed to contamination during EPMA analyses.”

Lines 52-54 and lines 326-329: How did the authors calculate H concentrations from a volume (of bubbles or cracks just from 2D photos)? What methods did they use? What

values did they take for the densities involved?

The H content in H-rich Fe alloy was obtained by volume expansion of the unit-cell measured by XRD analyses as mentioned above. Then, that in coexisting S-rich (H-poor) alloy was estimated from the relative proportion of bubbles and cracks in 2D images with respect to that in neighboring H-rich alloy. We just use the relative proportions between the H-rich and S-rich alloys such that densities are not involved in our estimates.

Okuchi (1997, Science) estimated H concentrations in Fe alloys from the volume of bubbles estimated in 2D images and reported the metal/silicate partition coefficient of H. Such partition coefficients are consistent with those recently determined by Tagawa et al. (2021, Nature Commun.) who obtained the H content from the volume expansion in the way same as that employed in this study.

What are the errors on the H concentrations found by this method?

The uncertainties in the H content in S-rich (H-poor) liquid range from 0.2 to 0.4 wt% as shown in Table 1. It corresponds to relative errors of ± 25 –100%, which is much larger than the ± 9 –20% errors in H concentrations in H-rich liquids that are estimated from the volume expansion of iron lattice.

Are the percentages written (10-20% and 70-80%) wt%, mol% or vol%?

These are the ratios of the areas of bubbles/cracks in S-rich (H-poor) liquids to those in H-rich liquids. We have rewritten and augmented the relevant descriptions in Lines 237–241;

“We assumed that the relative proportions of bubbles and cracks in neighboring H-rich and S-rich (H-poor) liquids represent the difference in the hydrogen content between them, following ref. 22. It gives the amounts of hydrogen in S-rich liquids to be 10–20% and 70–80% of those in coexisting H-rich liquids in Si/O-free and -bearing samples, respectively.”

3. Lines 114-117: What would be the consequences with regard to the silicates present in the magma ocean?

In response to this comment, we consider the consequence of liquid immiscibility during core segregation in a quantitative way. With a single-stage core formation model, the amount of S left in the magma ocean (MO) after metal segregation is determined by the metal/silicate partition coefficient of S, D_S and the masses of metal and silicate that are involved in chemical reaction. If core-forming metal was separated into S-rich and H-rich liquids, only the S-rich one included S and was involved in S partitioning. We calculate $X'_S(k)$ below to show how much the S content left in the MO increases as a function of k , a fraction of S-rich immiscible liquid in core-forming metal ($k=1$ when immiscibility does not occur);

$$X'_S(k) = \frac{X_S^{MO}(k)}{X_S^{MO}(k=1)} = \frac{1 + \frac{D_S M_{metal}}{M_{silicate}}}{1 + k \frac{D_S M_{metal}}{M_{silicate}}} \quad (\text{R1})$$

where $X_S^{MO}(k)$ represents S concentration in the MO after core segregation, and M_{metal} and $M_{silicate}$ are the masses of metal and silicate that reached chemical equilibrium, respectively. **Fig. R3** below illustrates the variations in $X'_S(k)$ for different $D_S M_{metal}/M_{silicate}$.

Fig R3. The S content left in the MO after core formation as a function of k (a fraction of S-rich liquid in core-forming metal) normalized by the S content when $k=1$ (immiscibility does not occur).

When considering a single-stage core formation at ~ 50 GPa and $\sim 3,500$ K (e.g. Siebert et al. 2012, EPSL), D_S is in the order of 10–100 (Suer et al. 2017, EPSL; Rose-Weston et al. 2009, GCA) and $M_{metal}/M_{silicate} = 0.5$. Therefore, $D_S M_{metal}/M_{silicate}$ is also in the order of 10–100, leading to $X'_S(k) \sim 2$ with $k=0.5$ (core-forming metals are half S-rich and half H-rich) (see Fig. R3 above). It means that ~ 2 times as much as S is left in the MO due to the liquid immiscibility. This supports the likelihood of FeS segregation from partially solidified MO after the core formation known as the “Hadean matte” scenario (O’Neill 1991, GCA).

The metal-silicate partition coefficients of some highly siderophile elements depend on S concentration in metal (Laurenz et al. 2016, GCA). Rubie et al. (2016, Science) argued that the Hadean matte is a possible explanation for suprachondritic Pd/Ir and Ru/Ir ratios in the present-day mantle. Some chalcophile elements like Cu and Pb may have been removed from the mantle by this mechanism as well (Savage et al. 2015, GPL; Wood & Halliday 2005, Nature). Our result supports these arguments.

We added the following sentence in Lines 122–123;

“If core-forming metal separated into half S-rich and half H-rich liquids, the amount of sulfur left in silicate melt could be doubled.”

Would these consequences be different if the Fe-S-H immiscibility appears once the core is formed and not during its segregation?

A multi-stage core formation model with heterogeneous accretion by Suer et al. (2017, EPSL) argued that mantle sulfur concentration can be explained without “Hadean matte” scenario (FeS segregation) if sulfur was delivered only in the last ~20% of accretion. Even in that case, however, if hydrogen was supplied together with sulfur (it is indeed very likely), it can cause the Fe-S-H liquid immiscibility in the late stages of core formation under 30–50 GPa considering relatively low-temperature *P-T* evolution paths for metal-silicate equilibrium.

In response to this comment, we have augmented the discussion on FeS segregation in Line 126–130;

“A recent model³⁹ of heterogeneous Earth accretion and multi-stage core formation argued that the mantle sulfur concentration is explained without sulfide segregation if sulfur was supplied only in the late stage of planet growth. However, it is likely that hydrogen was transported along with sulfur¹⁷, possibly leading to the Fe-S-H liquid immiscibility and the sulfide segregation.”

4. Lines 282: “The heating was limited to 3-10 sec...”. This heating time is extremely short. It would be important to justify that it is enough for each element (H, Si, C or S) by a calculation based on the two references mentioned for example.

Based on Helffrich (2014, EPSL), the diffusivities of S and Si in liquid iron are about 10^{-9} m²/s at CMB conditions and faster at lower pressures (those of H, C and O are higher). It requires only 0.1 sec for these elements to diffuse over a liquid pool with ~10 μm size in our experiments. Homogeneous liquid composition ensures the equilibrium miscible or immiscible liquid state. In addition, previous time-series experiments on Fe-S alloys demonstrated that coexisting liquid and solid compositions did not change after 1 sec (Mori et al., 2017, EPSL).

In response, we augmented the relevant statement in Line 192–196;

“The heating duration was limited to 3–10 sec to avoid temperature fluctuations. It is long enough for all the light elements examined here to diffuse over a ~10 μm size liquid pool⁴⁷, which ensures the equilibrium miscible and immiscible liquid states. Previous time-series melting experiments⁵⁵ on Fe-S alloys showed that results did not change with increasing heating duration from 1 to 120 sec.”

5. There are obviously significant temperature gradients throughout the experiments that could have prevented (at least chemical) equilibrium. In other words, can the calculated contents of H, C, S, and Si obtained in this study be used quantitatively as a starting point for modeling for the Earth and Martian core?

Even with a temperature gradient in a sample, local chemical equilibrium should attain, in particular between neighboring immiscible two liquids, because mixing/separation should occur instantaneously. The miscible and immiscible liquid states observed in this study are certainly applicable to the modeling of planetary interiors.

6. Extended Data Fig 2 b and c : I do not see the border between the two "liquids" where the authors drew it with the dotted lines...

The boundary between two immiscible liquids is clearly seen in a scanning ion microscope (SIM) image (see below). In all figures in Supplementary Fig. 1, contrasts in the S content are clear in both X-ray elemental map and the SIM image.

In response to this comment, we added a SIM image in Supplementary Fig. 1b.

SIM image of run #10 (Supplementary Fig. 1b)

On picture 2b: I see on the left and right edges a contrast identical to the one above.

See the newly added SIM image shown above.

On photo 2c: I see no difference in the contrasts (maybe it is due to the quality of my pdf file).

As the Reviewer pointed out, the contrast in O concentration between immiscible liquids is small, simply because their O contents are minor (<1 wt%).

7. Lines 347: I don't see the spherical shape or infiltration of Al₂O₃ grains into metal (photo 1e).

“Spherical shape” denotes the metal shown Fig. 1b. “Infiltration of Al₂O₃ grains” can be seen in Fig. 1e. We made these points clear in the revised manuscript.

We thank the Reviewer for his/her comments, which helped us to improve the Discussion and Methods sections.

Response to comments by Reviewer 3 (reviewer comments in bold and our response in blue):

The manuscript report experimental results on the immiscibility of the Fe-S-H system to Mbar pressures. The results were used for the discussions on the formation and termination of the Martian magnetic field, as well as the nature of the low-velocity layer below the core-mantle boundary in Earth's outer core. The experiments are certainly very challenging, especially for experiments approaching the Earth's core conditions. That is the reason why there are so few data points available. I find the implications of the data intriguing, but the conclusions are based on the sparse data. In particular, the discussion on the shallow part of the Earth's outer core (or E' layer) is speculative. Therefore, I do not recommend the publication of the manuscript in its present form.

On the basis of this comment, we have performed four additional experiments at relatively high pressures. As a consequence, the P - T locations of the miscible-immiscible boundary has been tightly constrained, in particular under the Earth's uppermost outer core conditions. See our detailed response below.

Main issues:

- The miscible/immiscible boundary was fitted from the very limited data. There supposed to be huge uncertainties in the fitted parameters (Line 78-81). In Fig. 2, dashed lines below and above the immiscible boundary (black line) were not described. I suppose these are confidence bands of the fittings.

As the Reviewer supposed, the dashed lines give the confidence band for miscible-immiscible boundary. We now mention it in the Fig. 2a caption.

- The constraints for the boundary are mainly from the three data points data below 40 GPa and the higher pressures data points only provide the lower bound. It is questionable how one can reliably fit the data to constrain the miscible/immiscible boundary up to the earth's core pressures. It will make a convincing case, if the authors can provide a few more data points at > 40-120 GPa where the Fe-S-H liquids is miscible.

Following this advice, we have conducted four additional experiments at 44–99 GPa and above 3,000 K (runs #12–15). Accordingly, the P - T location of the miscible-immiscible boundary has been tightly constrained, indicating more confidently that liquid immiscibility can occur in both the Martian and Earth's cores as well as at conditions for metal segregation during the Earth's core formation. See the revised Fig. 2a below (newly added experimental results are emphasized by red squares only in the figure below).

Fig. 2a. Experimental results showing a homogeneous single liquid (filled) and two immiscible liquids (open) in Fe-S-H (rectangles) and Fe-S-H-C/Si/O (triangles). Elements other than Fe, S and H are indicated. Miscible/immiscible boundary (black line) and its uncertainty band (dashed lines) are based on the P - T conditions of miscible/immiscible border in a sample (diamonds) (runs #1, 2 and 4) and those of miscible/immiscible liquids obtained near the boundary (runs #7 and 13–15).

Minor comments:

Line 48: It is better to specify the carbon content for the liquids.

Following this comment, we now mention the C contents in the revised manuscript.

Line 54-55: It is quite vague to state “relatively high temperatures” here. What temperature is considered high temperature?

In response to this comment, we now clarified as “>2,000 K at ~20 GPa and >~3,000 K at higher pressures”.

Line 338-340: Please describe in details the analysis of carbon in the Fe alloy samples, as we know it can be tricky and challenging to measure carbon by EPMA.

In response to this comment, we have augmented descriptions on the present EPMA analyses in Lines 248–253 as follows;

“Quantitative chemical analyses of Fe, S, Si, O and C were then performed by a field-emission-type electron probe microanalyzer (FE-EPMA, JEOL JXA-8530F) with a voltage of 12 kV and a beam current of 15 nA. No coating material was necessary. We used LIF (Fe), PET (S), TAP (Si), LDE1 (O) and LDE2H (C) as analyzing crystals, and Fe, pyrite, silicon, corundum and Fe₃C as standards. The X-ray counting time for peak/background was 20 sec/10 sec. The ZAF correction was applied.”

Table 1: How was the hydrogen content of immiscible liquids determined for run #3, 6, 7-11?

We found immiscible H-rich and S-rich (H-poor) liquids in these experiments. The H content in the H-rich liquid was estimated from the volume expansion of FeH_x with respect to that of Fe. The XRD peaks of FeH_x should have derived from the quenched H-rich liquids because they indicate $x \sim 1$. Subsequently we estimated the “ratio” of H concentration between the neighboring H-rich and -poor liquids from their relative proportions of bubbles and cracks in cross sections, following the method by Okuchi (1997 Science). The H content in the H-poor liquid was thus obtained from the abundance ratio and the H content in the H-rich liquid.

Indeed, Okuchi (1997 Science) estimated H concentrations in Fe alloys from the volume of bubbles estimated in 2D images and reported the metal/silicate partition coefficient of H. Such partition coefficients are consistent with those recently determined by Tagawa et al. (2021 Nature Commun.) who obtained the H content from the volume expansion in the way same as that employed in this study.

In response to this comment, we have augmented the relevant descriptions in Lines 237–239;

“We assumed that the relative proportions of bubbles and cracks in neighboring H-rich and S-rich (H-poor) liquids represent the difference in the hydrogen content between them, following ref. 22.”

Fig. 1d: the texture analysis of run #6 needs more description. Why does it appear that there are two domains in the miscible liquid: one is enriched in H and the other depleted in H?

The Reviewer is concerned that we can find two domains in a single miscible liquid. It is because there is a void next to a quenched liquid. It is likely that it was filled with hydrogen escaped from iron upon decompression (or possibly occupied by diamond formed from paraffin). The dark domain is marked with “void by H”.

What are those black domains between the two liquid pockets?

They consists of the Al_2O_3 pressure medium, carbon (diamonds) and unidentified hydrocarbons (or unreacted paraffins). In response to this comment, we now mention it in the caption of Fig. 1.

We thank the Reviewer for his/her advice on necessary additional experiments. These additional experiments greatly improved the manuscript.

REVIEWERS' COMMENTS

Reviewer #2 (Remarks to the Author):

In general, the authors have responded convincingly to my remarks. I would like to acknowledge the hard work and effort that the authors have made in responding point by point to my remarks (as well as to other reviewers). The additional figures and calculations they have made allow them to better present and argue their results and conclusions. I now strongly recommend this study for publication in your journal.

Valerie Malavergne

Reviewer #3 (Remarks to the Author):

The author has addressed the main concerns raised in the first round of review and provided experimental data from four additional measurements at 44-99 GPa. The experimental constraint on the miscible/immiscible boundary of the Fe-S-H liquids is much more reasonable in the revised Fig. 2a and relevant data. The additional work provides more convincing results support the main conclusion of the manuscript.

Some additional comments:

- Line 79-81: Here the authors suggest liquid Fe alloys can be classified into two groups, Fe-S-Si-O and Fe-H-C. This may worth some discussion in terms of interstitial (e.g. H or C) or substitutional (e.g., Si) solution of the light elements in the liquid Fe alloys. The discussion will inform some new understanding on the liquid structure of Fe-light-element alloys.

- Fig. 3: It may be helpful to state the cartoons for Earth and Mars are not up to scale.

Response to comments by Reviewer #2 (reviewer comments in bold and our response in blue):

In general, the authors have responded convincingly to my remarks. I would like to acknowledge the hard work and effort that the authors have made in responding point by point to my remarks (as well as to other reviewers). The additional figures and calculations they have made allow them to better present and argue their results and conclusions. I now strongly recommend this study for publication in your journal.

We thank the Reviewer for acknowledging our works.

Response to comments by Reviewer 3 (reviewer comments in bold and our response in blue):

The author has addressed the main concerns raised in the first round of review and provided experimental data from four additional measurements at 44-99 GPa. The experimental constraint on the miscible/immiscible boundary of the Fe-S-H liquids is much more reasonable in the revised Fig. 2a and relevant data. The additional work provides more convincing results support the main conclusion of the manuscript.

Some additional comments:

- Line 79-81: Here the authors suggest liquid Fe alloys can be classified into two groups, Fe-S-Si-O and Fe-H-C. This may worth some discussion in terms of interstitial (e.g. H or C) or substitutional (e.g., Si) solution of the light elements in the liquid Fe alloys. The discussion will inform some new understanding on the liquid structure of Fe-light-element alloys.

Following the Reviewer's comment, we have added two sentences about the mechanism of incorporating the light elements in Line 82–84 as follows;

“Such affinity might originate from the mechanism of incorporating light elements in liquid Fe; hydrogen and carbon have smaller atomic radii and therefore stronger interstitial characters than sulfur, silicon and oxygen (ref.18).”

- Fig. 3: It may be helpful to state the cartoons for Earth and Mars are not up to scale.

We now mention it in the caption of Figure 3.

We thank the Reviewer for favorable comments and additional advice.